# Investigation on Compressive Characteristics of Steel-Slag Concrete

**DOI:** 10.3390/ma13081928

**Published:** 2020-04-19

**Authors:** Thi-Thuy-Hang Nguyen, Duc-Hung Phan, Hong-Ha Mai, Duy-Liem Nguyen

**Affiliations:** 1Faculty of Civil Engineering, Ho Chi Minh City University of Technology and Education, 01 Vo Van Ngan St, Thu Duc District, Ho Chi Minh City 700000, Vietnam; hangntt@hcmute.edu.vn (T.-T.-H.N.); hungpd@hcmute.edu.vn (D.-H.P.); 2Faculty of Transportation Engineering, Ho Chi Minh City University of Transport, 02 Vo Oanh St, Ward 25, Binh Thanh District, Ho Chi Minh City 700000, Vietnam; ha.mh@ut.edu.vn

**Keywords:** steel-slag, size effect, brittleness, modulus of elasticity, Poisson’s ratio

## Abstract

The compressive characteristics of the steel-slag concrete were investigated through an experimental test. The term “steel-slag concrete” in this research work was defined as a kind of concrete using steel-slag material as a coarse aggregate replacement. Three types of the steel-slag concretes were examined under compression as follows: XT01, XT02, XT03 with their cement/water ratios of 1.76, 2.00, 2.21, respectively. The coarse aggregate used in producing concrete was steel-slag material, while the fine aggregate was traditional river sand; the ratio of coarse aggregate to fine aggregate was kept constant at a value of 1.98. Firstly, the age-dependent compressive strength of the steel-slag concretes were investigated up to one year; it was clear that the concrete strength increased rapidly in 7 days, then more and more slowly after that. Secondly, the modulus of elasticity and Poisson’s ratio of the steel-slag concretes were explored at the 28-day age. Thirdly, there was an important size and shape effect on the compressive strength of the XT02, and its significance of brittleness in failure was analytically analyzed. Lastly, the effects of water amount added in the XT02 on its compressive strength and slump were evaluated at the 28-day age.

## 1. Introduction

Industrial wastes and by-products have become a serious issue related to the environment, therefore, recycling them has really been attractive to researchers, owing to clear benefits: reducing construction cost, reducing negative environment impact, and serving natural energy and resources. Steel-slag is an industrial waste generated from steel manufacturing, specifically, as one ton of steel is produced, about 160 kg of steel-slag will be created [1]. A great volume of steel-slag in many countries is really in excess, becomes a pollution risk for environment and thus, requires more consumption of steel-slag as a recycled material. The useful application of steel-slag was reported in many fields. Pasetto et al. [2] and Poulikakos et al. [3] proposed to use steel-slag in road construction while Galán-Arboledas et al. [4] informed the utilization of steel-slag as ceramic building materials. Besides, Kourounis et al. [5] and Zhang et al. [6] mentioned that cement could be blended with steel-slag, which played its role as a mineral admixture for making concrete. Li et al. [7] explored that steel-slag powder possibly produced its hydraulic properties, affecting the hydration process like cement. Peng et al. [8] stated that the steel-slag powder contributed to better workability and viscosity of concrete, and, it could also produce a reduction in promptly autogenous shrinkage and adiabatic temperature growth [9,10]. Song et al. [11] and Nguyen et al. [12] recently reported that the mechanical resistance and self-sensing capacity of high-performance fiber-reinforced concrete could be enhanced using steel-slag powder as partial cement replacement material. Nonetheless, the use of steel-slag powder as a cement replacement material is still limited in quantity, because a large content of steel-slag powder in concrete mixtures may cause a negative effect on compressive resistance, the sulfate attack resistance of concrete, carbonation resistance and chloride–ion penetration resistance [13,14]. In the bigger particle form, steel-slag was also reported to be possibly used as a coarse aggregate or a fine aggregate in concrete composition [15,16]; this was because the steel-slag aggregate had better physical properties in comparison with the crushed limestone aggregate [17]. Concrete made from steel-slag aggregate could be denser than plain cement concrete, which leads to higher performance in mechanical strengths [18,19,20].

In Vietnam, there is a high and urgent demand in recycling steel-slag, owing to huge quantities in the steel manufacturing industry, e.g., only in Ba Ria-Vung Tau province in southern Vietnam, the annual production capacity from thirteen steelmaking plants was about 3.75 million tons of billet accompanied with steel-slag amount up to 412,000–562,000 tons per year [21]. Up to now, there are some available references [22,23,24,25,26,27] regarding the concrete containing Vietnamese steel-slag as an aggregate (called steel-slag concrete): Lam et al. [22,23,24] suggested to apply the steel-slag aggregate for roller-compacted concrete pavement, while Nguyen et al. [25,26] and Tran et al. [27] studied steel-slag aggregate for making concrete used in structural members. Although these references provided some information on steel-slag concrete, further investigation is still required as follows: 

(a) The compressive strength is always the key parameter of normal concrete and/or high-performance concrete. This is due to their strengths under compression being much greater than those under tension, even for strain-hardening fiber-reinforced concrete with its high tensile strength more than 10 MPa [28,29,30]. On the other hand, the compression strength of concrete has been well known to depend upon time during cement hydration, and clear information on compressive strength of steel-slag concrete will help better utilization of this concrete type, e.g., how significant is 28-day compressive strength in comparison with one-year compressive strength? In general, time is the main influencing factor affecting the construction progress.

(b) The scale effect of traditional concrete has been investigated for decades: the mechanical properties of traditional concrete were highly size and shape-dependent due to its brittle nature [31,32,33,34]. In compression, there have been two main different testing standards for compressive specimens using cubic shape or cylindrical shape. The cylindrical shape, with a size of 150 mm in diameter and 300 mm in height, has been applied mostly in Canada, United States, and Australia, whereas the cubic shape with dimensions of 150 mm or 100 mm have been applied mostly in Europe and Vietnam [35]. The Vietnamese standard TCVN 8218:2009 [35] provides conversion factors for the compressive strength of traditional concrete with various specimen sizes and shapes. Nonetheless, the size and shape-dependent compressive strength of steel-slag concrete is still questionable, and it should be investigated and compared with that of traditional concrete using existing models/theories. The clear understanding of the size and shape effect on the strength of steel-slag concrete would help civil engineers properly design structural members using steel-slag concrete.

(c) For traditional concrete, Poisson’s ratio was reported to be from 0.15 to 0.2 for normal-weight (ACI 318 [36]). Is the value of Poisson’s ratio for steel-slag concrete similar to traditional concrete?

The above questions have motivated this investigation, with specific objectives listed as follows: (1) to discover the age-dependent compressive strength of the steel-slag concretes, (2) to investigate the modulus of elasticity and Poisson’s ratio of steel-slag concrete, (3) to explore size and shape-dependent compressive strength of the steel-slag concrete, and, (4) to investigate the effect of water amount added in mixture on compressive properties of steel-slag concrete. Except objective 1, the other objectives were studied with experimental tests at the 28-day age.

## 2. Experimental Work

### 2.1. Materials and Specimen Preparation

In the previous studies [25,26], the authors investigated the compositions of steel-slag concretes in a wide range of partial materials to achieve the best workability and mechanical strengths, and these properties were then compared with those of traditional concretes using natural aggregate. Based on the initial studies, this research would focus on the steel-slag concrete with compressive strength ranging from 30 to 45 MPa. The compositions of three types of steel-slag concretes were presented in Table 1 and their names were as follows: XT01, XT02, XT03 containing cement/water (C/W) ratios of 1.76, 2.00, 2.21, respectively (i.e., W/C ratios of 0.57, 0.50, 0.45, respectively). In all types, steel-slag material was designed as the coarse aggregate while the fine aggregate was river sand. Figure 1 presents the layout of the testing program in this study. It was noticed that there were two types of steel-slag from an electric arc furnace: black slag (electric arc furnace slag, EAFS) and white slag (ladle furnace slag, LFS) which were generated by remelting unalloyed and alloyed steel waste, respectively [37]. The steel-slag aggregate used in this study was black slag, as shown in Figure 2a. Figure 2b presents the photos of the prepared specimens. The size distribution of fine aggregate particles and coarse aggregate particles were displayed in Table 2. Table 3 supplies the physical properties of the steel-slag material used while Table 4 provides the chemical composition of steel-slag material. For the task of objective 1, the compressive specimens were tested using cube specimens of 100 mm. In the task of objective 2, the cylinder specimens with dimensions of 100 × 300 mm (diameter × height) were tested regarding ASTM C469 [38]. For objective 3, six types of sizes and shapes of the specimens would be prepared as follows: cube 70.7 mm, cube 100 mm, cube 150 mm, cylinder 70 mm, cylinder 100 mm and cylinder 150 mm in diameter; the cylinder height was designed to be double of the diameter. The 100 mm cube specimens were also used for objective 4. The cement type was used in this study was PCB40 INSEE, produced from Vietnam. 

A laboratory mixer with a volume of 150 liters was used to mix materials. All casted specimens were placed for 24 h in a laboratory with a temperature of 28 ± 5 °C prior to demolding, and after that, they were cured in water with temperature of 25 ± 5 °C. It was noticed that the temperature in laboratory and in the water tank mentioned above were very common in southern Vietnam, thus it was very suitable to practically apply the steel-slag concrete referring to the test results. The curing time depended upon the study objective. Next, the cured specimens were moved out of the water tank and dried at room temperature. For each series, at least three specimens were examined and then averaged for investigating the material properties. 

### 2.2. Test Setup and Procedure 

A MATEST machine with 150-ton capacity, as shown in Figure 3a, was used for the experimental tests. The machine was used by applying displacement-controlled loading under speed of 1.0 mm/min for all specimens. It was noticed that the cylinder compressive specimens should have their surfaces smoothed carefully before testing. All tests were conducted at room temperature of 28 ± 5 °C and relative humidity of 70–80%. The compressive strength (named $f_{c}^{’}$) was given by Equation (1), as the ratio of maximum applied load ($P_{max}$) to the section area of the specimen section (A). Figure 3b presents the 150 mm-diameter cylinder with a frame attached according to ASTM C469, for measuring both axial and lateral displacement during testing.
(1)$$f_{c}^{’}=\frac{P_{max}}{A}$$

## 3. Test Results and Discussion

### 3.1. Dependence of Compressive Strength on Testing Age of Steel-Slag Concretes

Table 5 and Figure 4 show the compressive strengths ($f_{c}^{’}$) of the steel-slag concretes with dependence upon testing age (*t*). As shown in Figure 4, the compressive strengths, regardless of steel-slag concrete types, developed quickly at the early testing age, and more and more slowly at later testing age. This observation was completely consistent with normal concrete [36]. At any testing age, the order of steel-slag concrete in terms of compressive strength was not changed as follows: XT03 > XT02 > XT01, i.e., the compressive strength increased with increasing of the C/W. The compressive strength at 7 days and 28 days were about 55–66% and 69–73% of the compressive strength at 365 days, respectively. 

In order to predict the $f_{c}^{’}$ at any value of *t*, the relationship between $f_{c}^{’}$ and *t* should be mathematically modeled. Based on the shapes of testing performances on Figure 4, a hyperbolic curve was suggested to fit the testing curves using Equation (2). The suggested equation was completely satisfied with the boundary condition: as the testing age was zero, the compressive strength of steel-slag concrete would be zero, and, as the testing age reached infinity, the compressive strength would reach a constant value.
(2)$$f_{c}^{’}\left(t\right)=\frac{t}{at+b}$$

Here *a* and *b* are the constants characterizing material properties. To obtain the values of *a* and *b*, Equation (2) was changed into Equation (3).
(3)$$\frac{t}{f_{c}^{’}\left(t\right)}=at+b$$

Equation (3) was in linear relationship with a common expression of y=ax+b, where $y=\frac{t}{f_{c}^{’}\left(t\right)}$, x=t. Using the least square method with linear regression, the value of *a* and *b* can be obtained. Equation (4) and Figure 4a provide the results of linear regression analyses while Equation (5) and Figure 4b provide the analyzed relationships between compressive strength and testing age of three steel-slag concretes.
(4)$$\{\begin{array}{c} XT01:\;y=0.019x+0.026\\ XT02:\;y=0.017x+0.145\\ XT03:\;y=0.015x+0.121 \end{array}$$
(5)$$\{\begin{array}{c} XT01:f_{c}^{’}\left(t\right)=\frac{t}{0.019t+0.206}\\ XT02:f_{c}^{’}\left(t\right)=\frac{t}{0.017t+0.145}\\ XT03:f_{c}^{’}\left(t\right)=\frac{t}{0.015t+0.121} \end{array}$$

Figure 5 shows the photos describing a typical cracking behavior of the steel-slag concretes: the failure crack propagated across the steel-slag particles (coarse aggregate), and this may be attributed to the lower strength of coarse aggregate (e.g., steel-slag) in comparison with the mixture strength. The lower strength here of the coarse aggregate can be explained by the honeycomb microstructure of the steel-slag aggregate. The cracking feature observed in steel-slag concrete was similar to that of in high strength concrete using a natural coarse aggregate [39], as illustrated in Figure 6.

### 3.2. Modulus of Elasticity and Poisson’s Ratio of Steel-Slag Concrete

Figure 7 presents the response curves of compressive stress versus axial strain, together with compressive stress versus lateral strain of three steel-slag concretes XT01, XT02, XT03. As presented in Figure 7, at the beginning of the load, the slopes (also indicating stiffness of a material) of the axial responses were lower than those of the lateral responses ($\emptyset_{a}$ < $\emptyset_{b}$), i.e., the absolute value of lateral strain (negative form in expression) was smaller than axial strain (positive form in expression), and, the linear relationship was observed at low-stress part, about one-third of the peak. Beyond the linear zone, the slope of response curve was gentler and gentler, owing to microcracks occuring. At failure, lateral strain was higher than axial strain, regardless of the tested series. Table 6 supplies the compressive parameters including compressive strength ($f_{c}^{’}$), lateral strain capacity ($\varepsilon_{lat}$), axial strain capacity ($\varepsilon_{axi}$), Poisson’s ratio ($\nu_{c}$), modulus of elasticity ($E_{c}$) and the compressive toughness ($T_{c}$). The strain capacity was detected as the corresponding strain of the peak stress, and, the toughness was detected as the area below the curve of the compressive stress versus axial strain response within the peak. The comparative compressive parameters were presented in Figure 8. As shown in Figure 8, except the lateral strain capacity, all compressive parameters of the studied steel-slag concrete were increased as the C/W increased; the order of concrete types in term of compressive parameters as follows: XT03>XT02>XT01. In detail, XT01, XT02 and XT03 produced their parameters as follows: $f_{c}^{’}$ of 22.91, 32.26 and 35.68 MPa, respectively; $\varepsilon_{axi}$ of 2.60‰, 2.80‰ and 2.82 ‰, respectively; $\varepsilon_{lat}$ of 3.28‰, 4.70‰ and 3.48 ‰, respectively; $T_{c}$ of 46.92, 69.49 and 77.88 MPa.‰, respectively; $E_{c}$ of 31.12, 36.68 and 39.21 GPa, respectively; $\nu_{c}$ of 0.173, 0.180 and 0.196, respectively. 

Under uniaxial compression, the $\varepsilon_{axi}$ ranging between 2.60‰ and 2.82‰ of the steel-slag concretes were agreed with a strain range of (2‰–3‰) of traditional concrete. Besides, the $\varepsilon_{lat}$ of the steel-slag concretes were observed to be higher than the $\varepsilon_{axi}$, from 1.23 to 1.68 times. For common concrete, the toughness in compression is much higher than the toughness in tension or flexure, and this can be attributed as concrete being strong under compressive loading but weak under tensile loading. With $T_{c}$ ranging from 46.92 to 77.88 MPa.‰, the compressive toughness of the steel-slag concrete was lower than that of high-performance fiber-reinforced concrete about 1.5–3 times [34], and lower than that of ultra-high-performance fiber-reinforced concrete about 5–8 times [40]. Both Poisson’s ratio and modulus of elasticity (MPa) were material parameters characterizing elastic performance. The Poisson ratios of the steel-slag concretes, detected in a range from 0.173 to 0.196, were very suitable with that of traditional concrete, having a range of (0.15–0.25) [37]. According to ACI [36], modulus of elasticity of concrete could be related with compressive strength under square root proportionality, as given in Equation (6), $k_{E}$ = 0.043.
(6)$$E_{c}=k_{E}w_{c}^{1.5}f_{c}^{’0.5}$$

In Equation (6), $w_{c}$ was the unit weight of the concrete (kg/m^3^) and $f_{c}^{’}$ was the compressive strength (MPa) measured at 28-day age using cylinder of 150 × 300 mm (diameter × height). For three steel-slag concretes with a total of 11 specimens, the $k_{E}$ = 0.049 was experimentally drawn with an average value of $w_{c}$ = 2600 kg/m^3^.

### 3.3. Size and Shape-Dependent Compressive Strengh of the Steel-Slag Concrete

In this part, all specimens were tested using series XT02 at the testing age of 28 days. Table 7 summarizes the testing data of six series tested. Figure 9a displays the $P_{max}$ value while Figure 9b performs the $f_{c}^{’}$ value. There was a converse trend in these two figures: as the size of the specimen increased with the same shape, the $P_{max}$ increased but the $f_{c}^{’}$ decreased, and this was called the phenomenon of size effect often occurring in brittle or quasibrittle materials. Up to now, two primary approaches have been available for clarifying the size-dependent strength of quasibrittle materials as follows: the statistical approach and the deterministic approach [41,42]. Both approaches would be analyzed for the steel-slag concrete in the following.

#### 3.3.1. Statistical Approach about Size Effect on Compressive Strength of Steel-Slag Concrete

A representative theory for statistical approach of size effect is Weibull’s size effect law. According to this law, the strength of brittle material is not constant, it is size-dependent: strength of a small specimen would be comparatively higher than that of a large specimen. The statistical approach of size effect can be visibly explained by a model of chain under tension and a model of brick stack under compression, as illustrated in Figure 10 [34]. According to these models, the large specimens have more elements thus, have a higher probability of defect elements resulting in breakdown [32].

Equation (7) performs the relationship of two specimens having different effective volumes of $V_{E1}$ and $V_{E2}$, and their corresponding strengths of $S_{1}$ and $S_{2}$, respectively. In Equation (7), m is named Weibull modulus and considered as a material parameter informing a significance of size effect, i.e., the smaller of m value indicated the more significant size effect. Regarding Weibull’s law, the failure probability, $P_{f}\left(S\right)$, is performed in Equation (8) then translated into logarithm form, given by Equation (9).
(7)$$S_{1}/S_{1}={\left\{V_{E2}/V_{E1}\right\}}^{\frac{1}{m}}$$
(8)$$P_{f}\left(S\right)=1-exp\left[-V_{E}{\left(\frac{S}{S_{o}}\right)}^{m}\right]$$
(9)$$ln\left\{ln\left[\frac{1}{\left(1-P_{f}\left(S\right)\right)}\right]\right\}=mln\left(s\right)+\ln\left(V_{E}\right)-mln\left(S_{o}\right)$$

In the analysis, $P_{f}\left(S\right)$ = i/(n+1), n is the total of tested specimens used for analysis, i is strength order of $S_{1}\leq S_{2}\ldots \leq S_{i}\ldots \leq S_{n}$, $S_{o}$ is the scale parameter and S is the failure strength. Equation (9) is a linear function of y=ax+b, where $y=ln\left\{ln\left[\frac{1}{\left(1-P_{f}\left(S\right)\right)}\right]\right\}$, a=m, x=ln(S). Using the least square method with linear regression, the value of a and b can be obtained.

Figure 11a shows the attainment of Weibull modulus (m) for both cube specimen strength and cylinder specimen strength using the least-squares method. The analysis results were as follows: m= 6.8 using nine cube specimens and m= 11.32 using nine cylinder specimens. Figure 11b shows the Weibull distribution of compressive strength with the m values obtained. It was noticed that the larger value of m, obtained from the cylinder specimen, indicated that the cylinder specimen was less sensitive to size in comparison with the cube. Besides, the obtained m for compressive strength of the steel-slag concretes were entirely suitable, with Weibull modulus for strength of the traditional concrete having a range from 4.2 to 24.2 [43].

#### 3.3.2. Deterministic Approach about Size Effect on Compressive Strength of Steel-Slag Concrete

Bažant’s size effect law has representatively performed the deterministic approach for quasibrittle materials based on the theory of fracture mechanics [32]. According to this law, the phenomenon of size effect related to the release of stored energy into the crack front [44]. The crack propagation associated with the fracture process zone (FPZ) of the material, which was defined as a region ahead of a traction free crack tip concentrating a great stress/strain [45]. The crack called here may be an air pore or a flaw embedded inside material in manufacturing process. Under loading, stress within the FPZ was redistributed with plastic performance whereas stress beyond the FPZ was still elastic. Since the FPZ was dependent upon material and independent upon specimen size, the FPZ size fraction in specimen was not the same. Figure 12 describes the FPZ size in comparison with specimen size as follows: it was important in small specimen and unimportant in big specimen, and as a result, small specimen trended to follow strength criteria. 

Though Bažant’s size effect law was primarily established for specimens subjected to tensile and flexural loading, Kim et al. [46] proposed to use it for compressive loading. This can be attributed to failure of compressive specimen mostly related with local tensile cracks or shear cracks happening at defects inside the specimen. Equation (10) gives relationship between strength and specimen size according to Bažant’s size effect law.
(10)$$\sigma_{N}=Bf_{t}{\left(1+\frac{D}{D_{o}}\right)}^{-1/2}$$
where D is the specimen size, $f_{t}$ is the tensile strength, $\sigma_{N}$ is the studied strength, $D_{o}$ and B are the material parameters considered to be constant. Like Weibull modulus in the statistical approach, $D_{o}$ here characterizes the brittleness of material. The $\sigma_{N}$ in this research was applied for the compressive strength ($f_{c}^{’}$).

Figure 13 presents Bažant’s size effect law under logarithm profile for a quasibrittle material. As presented in Figure 13, a dashed horizontal line reveals strength criteria for plastic material; a dashed line with a slope of −1/2 reveals linear-elastic fracture mechanics (LEFM) criteria for pure linear-elastic failure response. The failure of a quasibrittle material according to Bažant’s size effect law is displayed by a solid curve. Clearly, this response curve moves toward the strength criteria with small-sized specimen and toward LEFM criteria with big-sized member. Equation (10) can be written into Equation (11), which is a linear function of y=ax+b, where $y={\left(1/\sigma_{N}\right)}^{2},$
x=D,
$a=1/\left[{\left(B.f_{t}\right)}^{2}D_{o}\right]$ and $b=1/{\left(B.f_{t}\right)}^{2}$. Applying the least square method for nine cube specimens and nine cylinder specimens, the linear functions of them were found and given by Equation (12). Equation (13) described Bažant’s size effect law using the steel-slag concrete with $D_{o}=38.16$ mm for cube specimen and $D_{o}=122.61$ mm for cylinder specimen. Since the cube specimen produced the lower $D_{o}$, the significance of size effect using cube specimen was greater than that using the cylinder specimen. Figure 14 presents the attainment of $D_{o}$ and the graphs of Bažant’s size effect law for the steel-slag concrete. The experimental data for both cube and cylinder specimen were in the transition between the plastic zone and LEFM zone, however, the measured data of the cube specimen were nearer the line of LEFM than the line of strength criterion.
(11)$${\left(\frac{1}{\sigma_{N}}\right)}^{2}=\frac{1}{{\left(B.f_{t}\right)}^{2}D_{o}}D+\frac{1}{{\left(B.f_{t}\right)}^{2}}$$
(12)$$\{\begin{array}{c} Cube:\;\;\;\;\;y=0.00000372x+0.0001419\\ Cylinder:\;\;\;y=0.00000355x+0.0004355 \end{array}$$
(13)$$\{\begin{array}{c} Cube:\;\;\;\;\;f_{c}^{’}=83.94{\left(1+\frac{D}{38.16}\right)}^{-1/2}\;\;\;\;\;\;\;\\ Cylinder:\;\;\;f_{c}^{’}\left(t\right)=47.92{\left(1+\frac{D}{122.61}\right)}^{-1/2} \end{array}$$

#### 3.3.3. Deriving Conversion Factors for the Compressive Specimens Using the Steel-Slag Concrete with Various Sizes and Shapes

There has been a great demand for converting compressive strength between various sizes and shapes of specimens. This is due to the fact that many standard codes are still available using different standard specimens. From the testing data measured in this research, the conversion factors (λ) of the steel-slag concrete (XT02 series) were given by Equation (14) using the 150 mm cube specimen as the control specimen. Table 8 presents the derived λ of the steel-slag concrete and the comparison with those of traditional concretes. Generally, the conversion factors of the steel-slag concrete were somewhat smaller than those of traditional concrete but their differences were comparatively small. This meant that the conversion factors of traditional concrete that can be applied for steel-slag concrete.
(14)$$\lambda =\sigma_{cu}^{CU150}/\sigma_{cu}^{CSpe.type}$$

Here, $\sigma_{cu}^{CU150}$ is the compressive strength of 150 mm cube specimen, which was assigned as the control specimen, and $\sigma_{cu}^{CSpe.type}$ is the compressive strength of the tested series.

### 3.4. Effect of the Added Water Amount on Compressive Strength of Steel-Slag Concrete

There is a strong correlation between C/W and compressive strength in addition to workability of concrete. More water than its requirement for completing hydration would cause a reduction in strength because of more water-filled pore spaces between the grains. On the contrary, less water than its requirement would not be enough for hydrating all cement in the mixture. In this section, the mixture XT02, presented in Table 2, was selected as the control mixture, and, the specimen type for testing was the 100 mm cube specimen. The cement amount in the mixture composition was fixed at 2.00 while the water amount was changed from 0.7 to 1.4, according to the weight ratio. 

Table 9 and Figure 15 present the effect of C/W on slump and compressive strength of the steel-slag concretes at 28-day age. It was observed that the slump curve decreased with increasing of the C/W while the compressive strength was different: it increased with increasing of the C/W up to 2.5 and decreased after that, i.e., the highest strength was at the C/W of 2.5 (C/W = 2.00/0.8). The relationship between compressive strength and the C/W of the steel-slag concretes were compared with traditional concrete using the bound zone reported by Kosmatka et al. [47], as displayed in Figure 16. Since the bound of traditional concrete was built using cylinder specimens of 150 × 300 mm, the compressive strengths of the tested specimens in this section were firstly translated into compressive strengths of the such cylinder. As shown in Figure 16, although two tested series were nearly on the lower bound, the tested specimens were generally within the bound, i.e., the steel-slag concretes were the same as the traditional concrete in terms of relationship between compressive strength and the C/W. Noticeably, the presence of free CaO and MgO in the steel-slag aggregate might result in an important volumetric expansion for concrete, leading to bad structural performances and damages [48], hence it is involved in more further studies on this problem to evaluate.

## 4. Conclusions

In this research, the compressive characteristics of the steel-slag concretes were investigated focusing on four objectives. Some conclusions could be drawn as follows:Regardless of testing age, the order of steel-slag concretes in term of compressive strength was not changed as follows: XT03 > XT02 > XT01. The compressive strength at 7 days and 28 days were about 55–66% and 69–73% of compressive strength at 365 days, respectively.The axial strain capacities of the steel-slag concretes were in a range between 2.60‰ and 2.82‰. The lateral strain capacities of the steel-slag concretes were higher than axial strain capacities from 1.23 to 1.68 times. The derived Poisson’s ratios and moduli of elasticity of the steel-slag concretes were usual in comparison with traditional concrete.The studied steel-slag concrete demonstrated clear size effects on compressive strength, in both cube specimens and cylinder specimens, i.e., the smaller-sized specimen would produce the higher compressive strength.According to Weibull’s size effect law, the obtained Weibull modulus for the compressive strength of the steel-slag concrete, m= 6.8 for cube specimen and m= 11.32 for cylinder specimen, were entirely in the range from 4.2 to 24.2, the zone of the Weibull modulus for strength of the traditional concrete.In Bažant’s size effect law, the obtained material parameter of the steel-slag concrete were as follows: $D_{o}=38.16$ mm for cube specimen and $D_{o}=122.61$ mm for cylinder specimen. The response curves describing Bažant’s size effect law of the steel-slag concrete were presented.The conversion factors for compressive strength using different sizes and shapes of the steel-slag concrete were explored. Generally, the conversion factors of the steel-slag concrete were slightly lower than those of traditional concrete, however, their differences were comparatively small.The slump value of the steel-slag concrete decreased with increasing of the cement/water ratio, while its compressive strength was different: it increased with increasing of cement/water up to 2.5 and decreased after that, the optimal cement/water was 2.5 to obtain the highest strength. The relationship between the compressive strength and cement/water ratio of the steel-slag concretes was almost inside the bound zone of traditional concrete.

## Figures and Tables

**Figure 1 materials-13-01928-f001:**
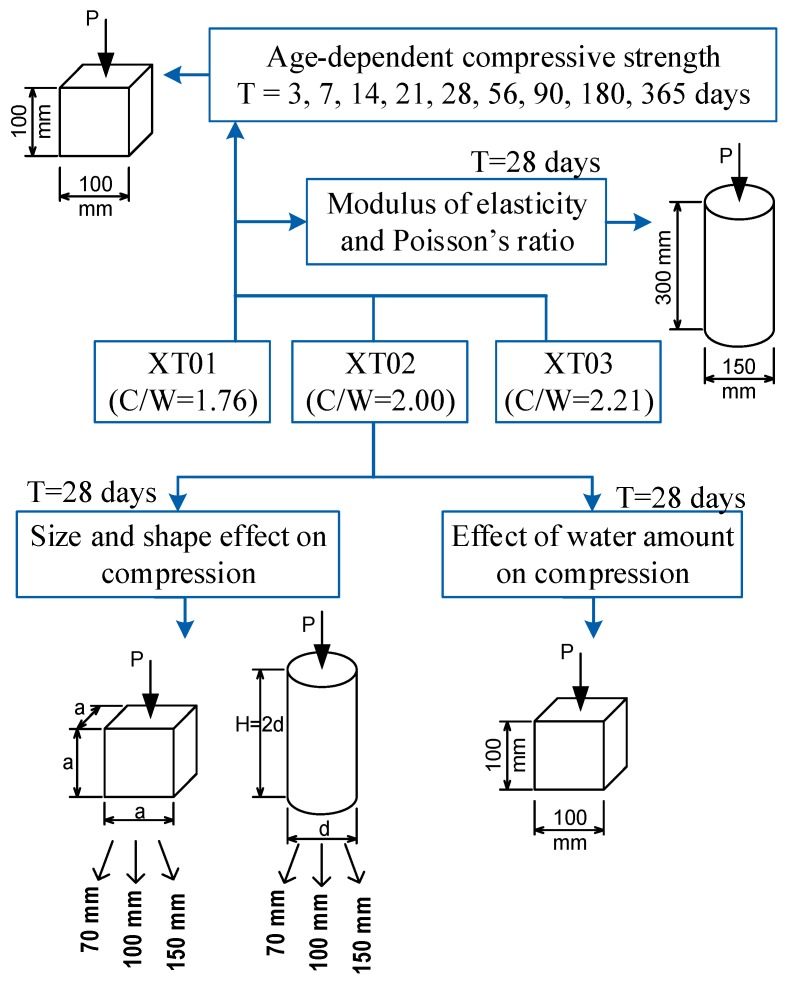
Layout of testing program.

**Figure 2 materials-13-01928-f002:**
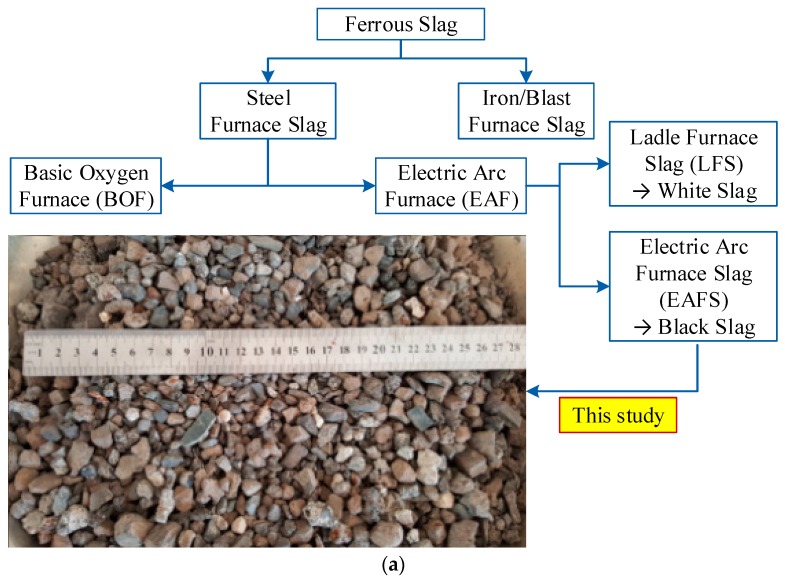
Photos of steel-slag and prepared specimens. (**a**) Steel-slag used as coarse aggregate. (**b**) Specimens with different sizes and shapes.

**Figure 3 materials-13-01928-f003:**
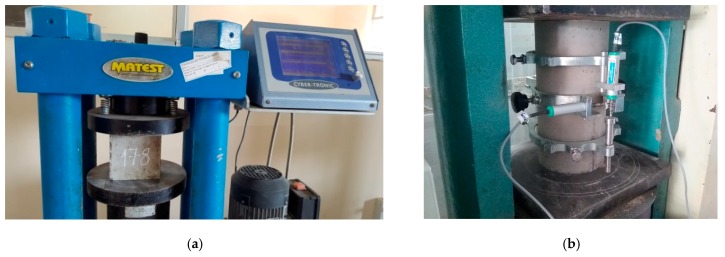
Photos of test setup. (**a**) Determining compressive strength. (**b**) Determining Poisson‘s ratio and modulus of elasticity.

**Figure 4 materials-13-01928-f004:**
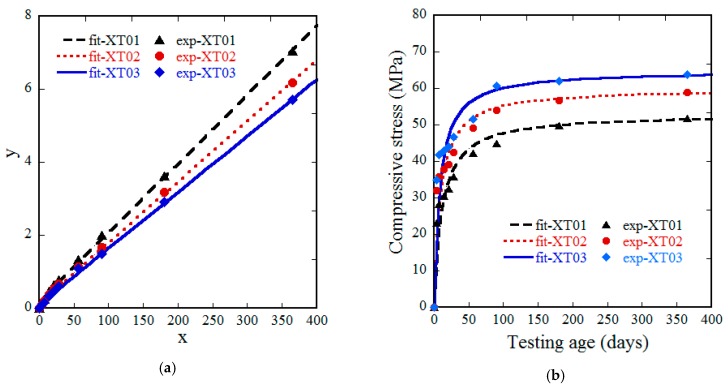
Relationship between compressive strength and testing age, according to regression analysis of three steel-slag concretes. (**a**) Linear least square regression. (**b**) Fitting curve using hyperbolic profile.

**Figure 5 materials-13-01928-f005:**
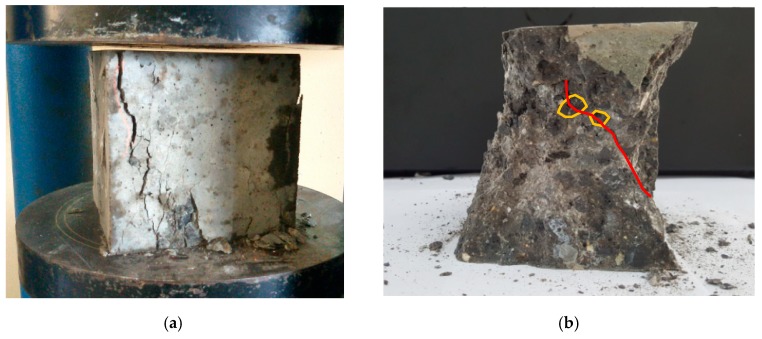
Typical crack under compression of 100 mm cube specimen of the steel-slag concrete. (**a**) Failure crack. (**b**) Cracking feature.

**Figure 6 materials-13-01928-f006:**
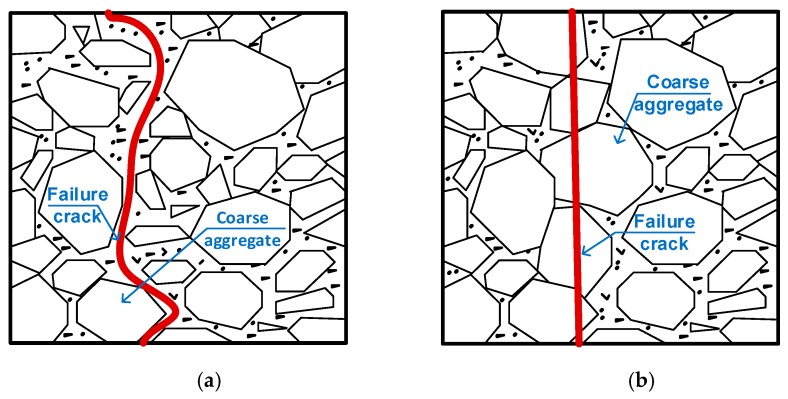
Failure cracking characteristics of normal concrete. (**a**) Ordinary strength concrete. (**b**) High strength concrete.

**Figure 7 materials-13-01928-f007:**
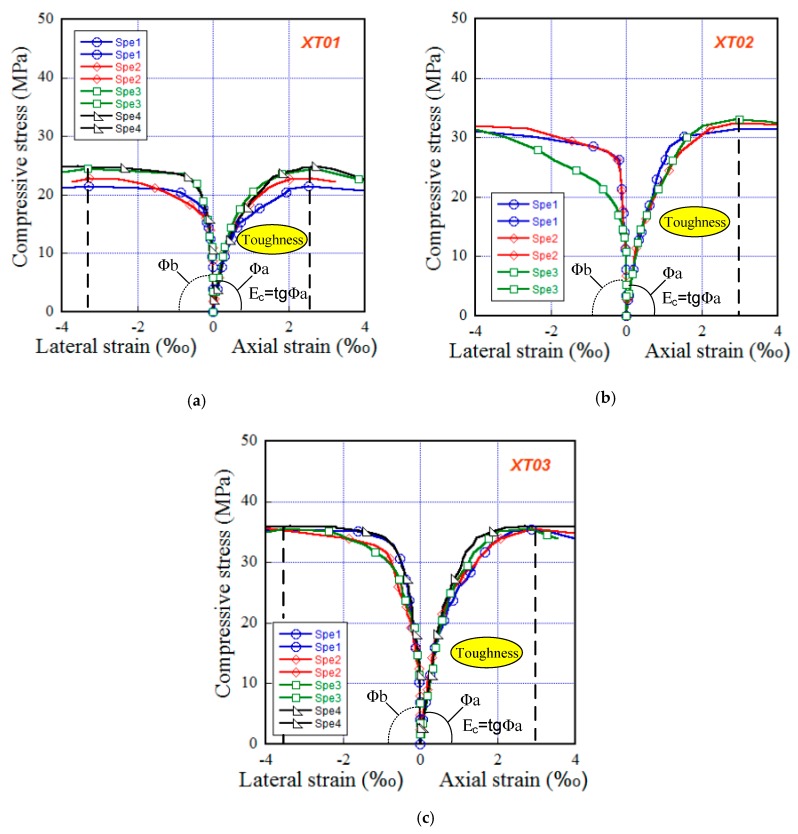
Compressive responses of the investigated steel-slag concretes at the 28-day age using cylinder specimens of 150 × 300 mm. (**a**) XT01. (**b**) XT02. (**c**) XT03.

**Figure 8 materials-13-01928-f008:**
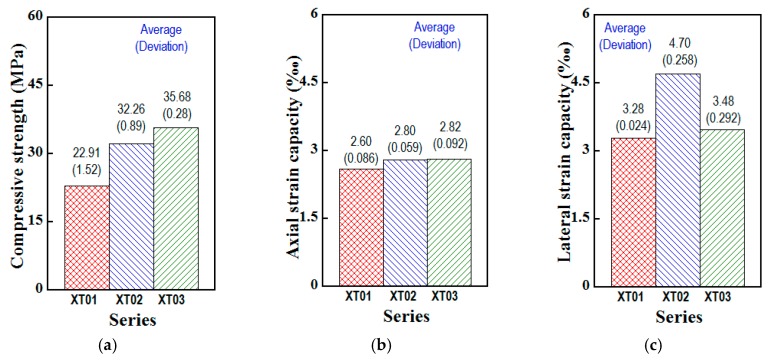
Comparison of compressive parameters at the 28-day age of three steel-slag concretes using cylinder specimens of 150 × 300 mm. (**a**) Compressive strength. (**b**) Axial strain capacity. (**c**) Lateral strain capacity. (**d**) Compressive toughness. (**e**) Modulus of elasticity. (**f**) Poisson’s ratio.

**Figure 9 materials-13-01928-f009:**
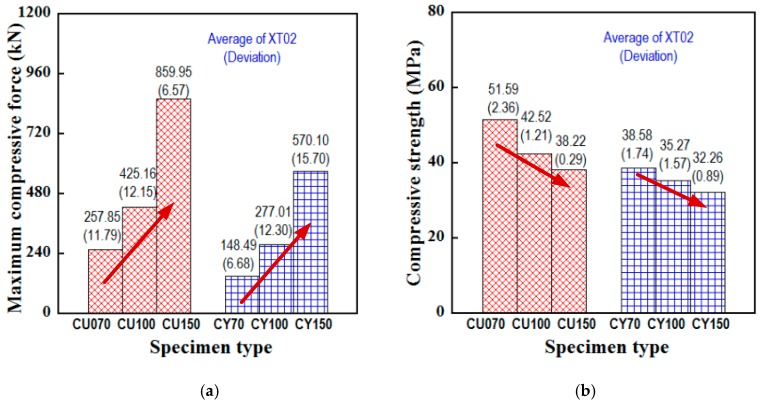
Dependence of compressive strength at the 28-day age upon specimen size and shape of the XT02. (**a**) Peak compressive force. (**b**) Peak compressive strength.

**Figure 10 materials-13-01928-f010:**
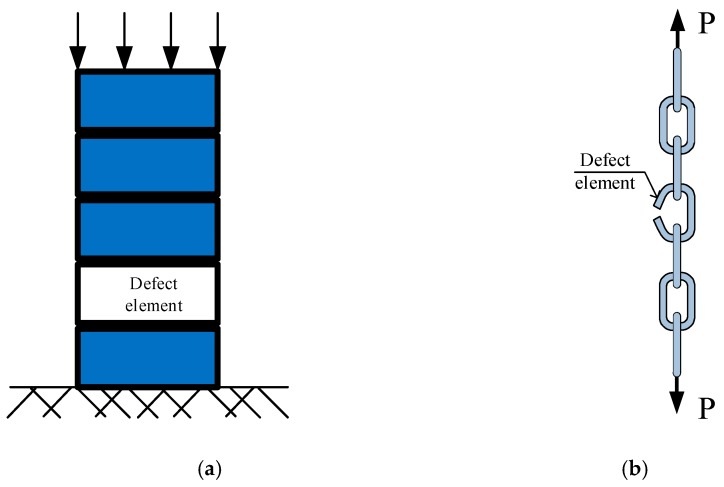
Models for explaining statistical approaches. (**a**) Brick stack under compression. (**b**) Chain under tension.

**Figure 11 materials-13-01928-f011:**
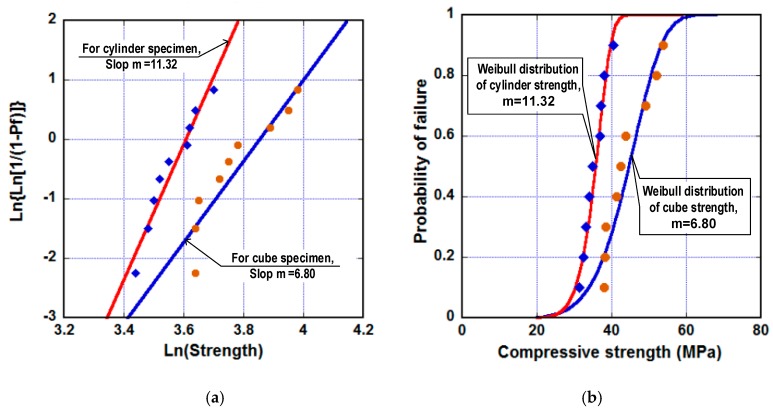
Applying Weibull’s size effect law to the steel-slag concrete. (**a**) Attaining Weibull modulus. (**b**) Weibull distribution of the compressive strength.

**Figure 12 materials-13-01928-f012:**
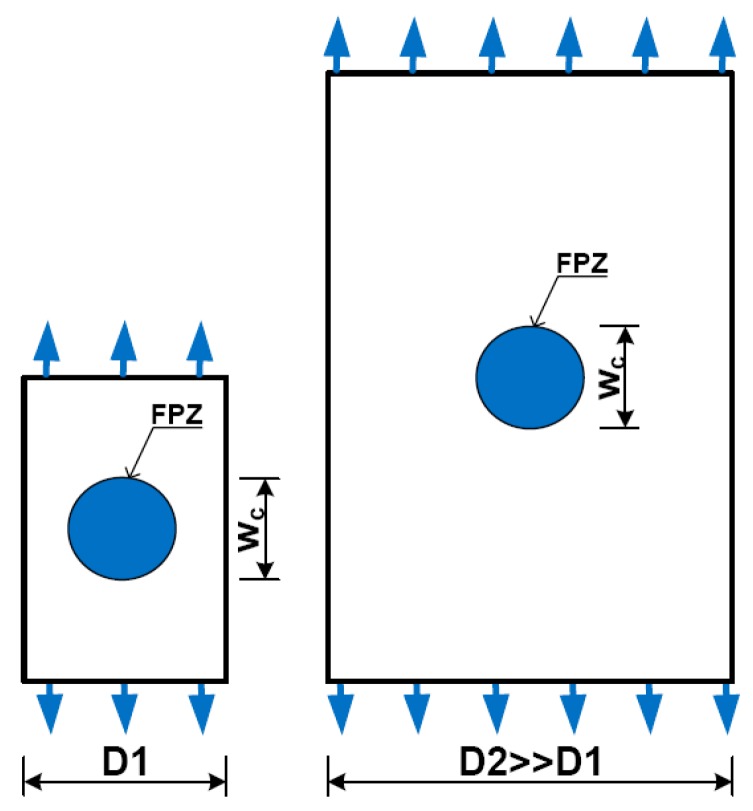
Describing FPZ size in comparison with specimen size.

**Figure 13 materials-13-01928-f013:**
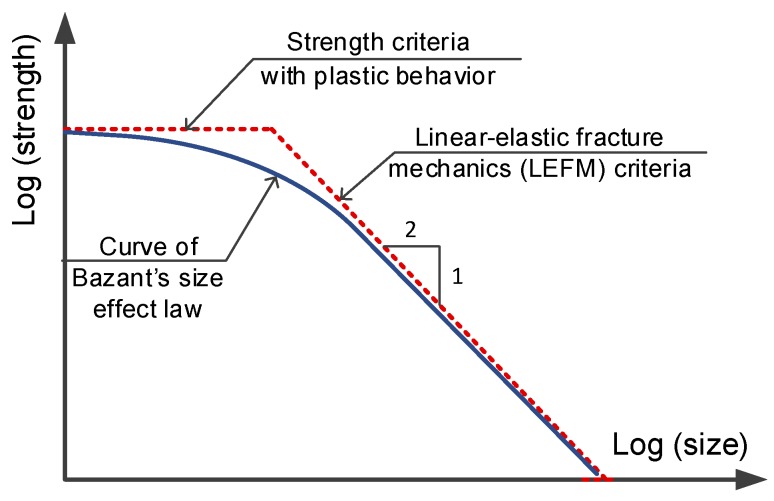
Describing Bažant’s size effect law on strength.

**Figure 14 materials-13-01928-f014:**
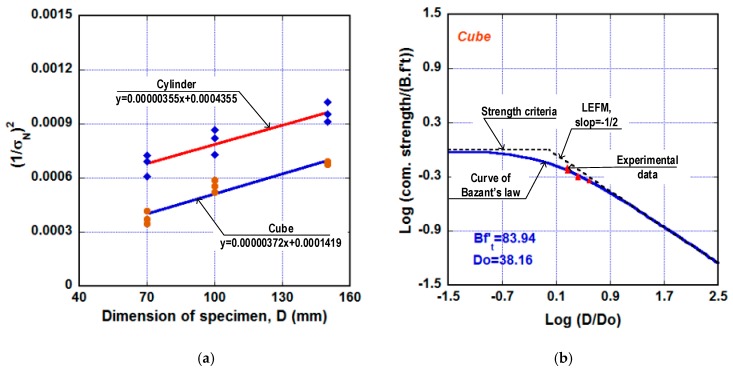
Applying Bažant’s size effect law for the steel-slag concrete. (**a**) Attaining of material parameters of Bažant’s size effect law. (**b**) Bažant’s size effect law for cube-shaped specimen. (**c**) Bažant’s size effect law for cylinder-shaped specimen.

**Figure 15 materials-13-01928-f015:**
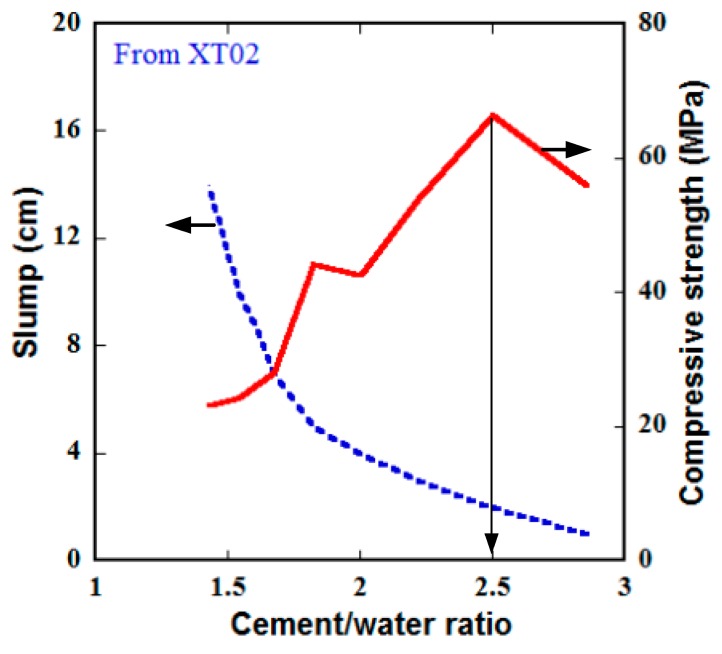
Effect of cement/water ratio on slump and compressive strength of the steel-slag concretes at the 28-day age using cube specimens of 100 × 100 × 100 mm.

**Figure 16 materials-13-01928-f016:**
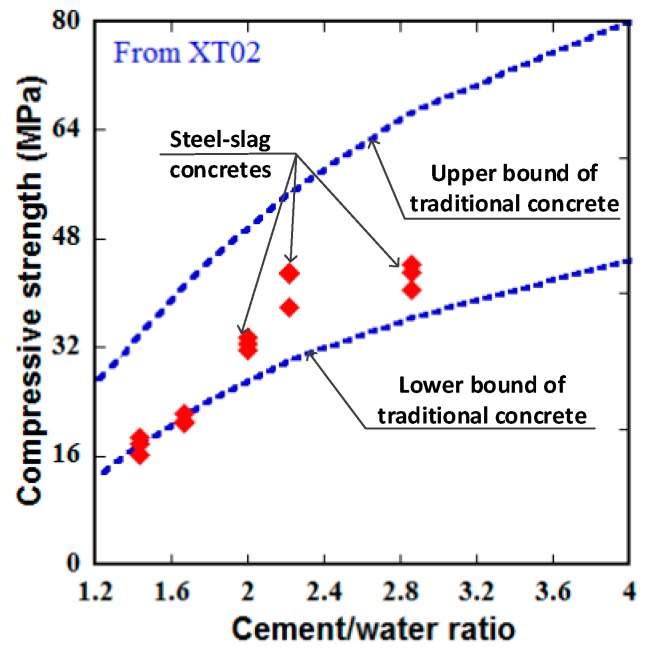
Relationship between 28-day compressive strength and cement/water ratio of the steel-slag concretes in comparison with traditional concrete using cylinder specimens of 150 × 300 mm.

**Table 1 materials-13-01928-t001:** Composition of steel-slag concretes.

Series	Cement	Fine Aggregate	Steel-Slag	Water	C/W (W/C)	Water Curing
XT01	1.76	3.39	6.72	1.00	1.76 (0.57)	25 ± 5 °C
XT02	2.00	3.30	6.55	1.00	2.00 (0.50)	
XT03	2.21	3.22	6.39	1.00	2.21 (0.45)	

**Table 2 materials-13-01928-t002:** Particle size distribution of fine aggregate and coarse aggregate.

Fine Aggregate (River Sand)		Coarse Aggregate (Steel-Slag)	
Size of Sieve (mm)	Pass (%)	Size of Sieve (mm)	Pass (%)
4.75	100	-	-
2.36	91.5	-	-
1.18	73.4	37.5	0
0.6	54	19	9
0.3	24.5	9.5	56
0.15	7	4.75	94.9
<0.14	0.0	19	100.0

**Table 3 materials-13-01928-t003:** Physical properties of steel-slag material.

Physical Properties	Steel-Slag	
D_min_-D_max_	5-20	mm
Specific gravity	3.56	g/cm^3^
Bulk dry specific gravity	3.32	g/cm^3^
Bulk saturated surface dry specific gravity	3.39	g/cm^3^
Water absorption	2.1	%
Bulk density	1720	kg/m^3^
Voids	48.2	%

**Table 4 materials-13-01928-t004:** Chemical composition of steel-slag material (%).

SiO_2_	Al_2_O_3_	FeO	Fe_2_O_3_	CaO	MgO	Na_2_O	K_2_O	TiO_2_	P_2_O_5_	SO_3_	Loss on Ignition
55.27	18.56	0.20	11.66	5.77	1.85	1.34	1.72	1.02	1.48	0.39	0.21

**Table 5 materials-13-01928-t005:** Compressive strength of steel-slag concrete at various ages.

Age (day)	Compressive Strength (MPa)		
	XT01	XT02	XT03
3	23.03	31.96	34.81
7	28.33	35.79	41.81
14	30.36	37.88	42.87
21	32.53	39.16	43.92
28	35.81	42.52	46.72
56	42.15	49.05	51.58
90	44.91	54.07	60.63
180	49.73	56.66	62.01
365	51.83	59.00	63.80

Note: These data use the cubic specimen of 100 mm.

**Table 6 materials-13-01928-t006:** Compressive parameters of the steel-slag concretes at 28-day age.

Series	Parameter	Maximum Force, $P_{max}\;(\mathrm{kN})$	Compressive Strength $f_{c}^{’}\;(\mathrm{MPa})$	Lateral Strain Capacity $\varepsilon_{lat}(‰)$	Axial Strain Capacity $\varepsilon_{axi}(‰)$	Poisson’s Ratio $\nu_{c}$	Modulus of Elasticity $E_{c}\;(\mathrm{GPa})$	Toughness $T_{c}$ (MPa. ‰)
XT01	Specimen1	378.30	21.41	−3.282	2.509	0.1705	30.35	40.94
	Specimen2	404.17	22.87	−3.260	2.610	0.1743	31.24	47.47
	Specimen3	432.00	24.45	−3.307	2.681	0.1726	31.78	52.35
	Specimen4	441.40	24.98	−3.578	2.693	0.1710	32.49	51.49
	Average value	413.97	22.91	−3.283	2.600	0.1725	31.12	46.92
	Standard deviation	26.86	1.52	0.024	0.086	0.002	0.72	5.73
XT02	Specimen1	553.70	31.33	−4.418	2.841	0.1753	36.50	71.45
	Specimen2	571.60	32.35	−4.756	2.737	0.1875	36.63	66.02
	Specimen3	585.00	33.10	-4.924	2.836	0.1766	36.91	71.00
	Average value	570.1	32.26	−4.699	2.805	0.1798	36.68	69.49
	Standard deviation	15.70	0.89	0.258	0.059	0.007	0.21	3.01
XT03	Specimen1	626.82	35.47	−3.193	2.882	0.1930	38.93	77.61
	Specimen2	628.69	35.58	−3.877	2.899	0.1926	39.06	79.86
	Specimen3	629.06	35.60	−3.504	2.807	0.1950	39.34	77.58
	Specimen4	637.73	36.09	−3.357	2.697	0.2028	39.51	76.46
	Average value	630.575	35.68	−3.483	2.821	0.196	39.21	77.88
	Standard deviation	4.87	0.28	−3.282	0.092	0.005	0.27	1.42

Note: $f_{c}^{’}$ is the compressive strength using cylindrical specimen with dimension of 150 × 300 mm.

**Table 7 materials-13-01928-t007:** Compressive resistances of the XT02 series at 28-day age.

Specimen Type (Specimen Name)	Parameter	Maximum Force, $P_{max}\;(\mathrm{kN})$	Strength $f_{c}^{’}\;(\mathrm{MPa})$
70.7 × 70.7 × 70.7 (CU070)	Specimen1	268.53	53.72
	Specimen2	245.20	49.05
	Specimen3	259.84	51.98
	Average value	257.85	51.59
	Standard deviation	11.79	2.36
100 × 100 × 100 (CU100)	Specimen1	437.41	43.74
	Specimen2	424.97	42.50
	Specimen3	413.11	41.31
	Average value	425.16	42.52
	Standard deviation	12.15	1.21
150 × 150 × 150 (CU150)	Specimen1	854.24	37.97
	Specimen2	858.49	38.15
	Specimen3	867.13	38.54
	Average value	859.95	38.22
	Standard deviation	6.57	0.29
Ø70 × 140 (CY70)	Specimen1	146.49	38.07
	Specimen2	155.94	40.52
	Specimen3	143.03	37.17
	Average value	148.49	38.58
	Standard deviation	6.68	1.74
Ø100 × 200 (CY100)	Specimen1	273.97	34.88
	Specimen2	266.52	33.93
	Specimen3	290.54	36.99
	Average value	277.01	35.27
	Standard deviation	12.30	1.57
Ø150 × 300 (CY150)	Specimen1	553.70	31.33
	Specimen2	571.60	32.35
	Specimen3	585.00	33.10
	Average value	570.10	32.26
	Standard deviation	15.70	0.89

**Table 8 materials-13-01928-t008:** Conversion factors in compressive strength using various specimen types.

Concrete Type	Basic Specimen	$\mathrm{Conversion}\;\mathrm{Factor},\;\lambda =\sigma_{cu}^{CU150}/\sigma_{cu}^{CSpe.type}$					
		CU070	CU100	CU150	CY70	CY100	CY150
Steel-slag concrete (XT02)	CU150	0.74	0.90	1.00	0.99	1.08	1.18
Traditional concrete [36]	CU150	0.85	0.91	1.00	1.16	1.17	1.20

**Table 9 materials-13-01928-t009:** Compressive strength and slump with various amount of water at the 28-day age.

C/W (W/C)	2.00/0.7 (0.35)	2.00/0.8 (0.40)	2.00/0.9 (045)	2.00/1.0 (0.50)	2.00/1.1 (0.55)	2.00/1.2 (0.60)	2.00/1.3 (0.65)	2.00/1.4 (0.70)
Slump (cm)	1	2	3	4	5	7	10	14
Compressive strength (MPa)	55.81	66.45	54.10	42.52	44.19	28.11	24.27	23.08

Note: Data using cubic specimen with size of 100 mm, the concrete with C/W = 2.00/1.00 was XT02.

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
