# Peer review of "Investigation on Compressive Characteristics of Steel-Slag Concrete"

_materials, 2020, doi:10.3390/ma13081928_

Round 1
Reviewer 1 Report
The paper "Investigation on Compressive Characteristics of Steel-Slag Concretes" is interesting and within the scope of Materials. I have some questions for the authors:
- in the title, I suggest using "concrete" instead of "concretes";
- the English language in the text should be revised: (e.g. line 61 "A Vietnamese standard named TCVN 8218:2009" should be modified by "The Vietnamese standard TCVN 8218:2009".
- revise the format of the manuscript (e.g. double blanks at line 29, red characters at lines 86, 123, 124, 126 and others)
- The scale of representation of Figure 2a lacks;
- please improve readibility of Figure 4
- The References format should be edited according to the journal’s style. Journals’ names must be in abbreviated format. The format for journal papers should be as: ‘Author 1, A.B.; Author 2, C.D. Title of the article. Abbreviated Journal Name Year, Volume, page range.’ For style of other kinds of documents, please check template document. Use N-dash for the page range, not hyphen. Include the digital object identifier (DOI) for all references where available. Some references are incomplete.
- line 411: what is the name of the Journal?
Author Response
Dear Sir/Madam,
Authors sincerely appreciate helpful comments from reviewer. The comments significantly improved our manuscript as shown in file attached.
Thank you very much for your consideration.

Reviewer 2 Report
The research work is very interesting, since it studies the behavior of the concrete in which the coarse aggregate is replaced by slag produced in the manufacture of steel, in order to be able to recycle this residue. The work is well structured, the results are presented clearly, and its analysis and conclusions respond to the results obtained.
Although the work is well done, it has some errors that must be corrected and are indicated below:
- Some text references are in red and in the bibliography, the references are disordered and not well referenced.
- Review table 2, there is a typo.
- Figure 9 is missing.
- In the title of Table 3 there is a typo.
- Review table 6, there is a typo.
In addition to these observations, some aspects need to be clarified in the work:
- There is talk of incorporating slag from steel, but when steel is manufactured, two slags are obtained: black slags and white slags. I understand that they refer to black slags in their work, but this aspect must be clarified in the document.
- Another important aspect is that the slags have volumetric stability problems. They should clarify the stabilization process carried out on them since their collection in the landfill, since it will influence the results obtained.
Author Response

(The authors gave the same response as above.)

Reviewer 3 Report
The aim of the article was to investigate the age‐dependent compressive strength of the steel‐slag concretes, the modulus of elasticity and Poisson’s ratio of steel‐slag concrete, to explore size and shape‐dependent compressive strength of the steel‐slag concrete and to investigate the effect of water amount added in mixture on compressive properties of this kind of concrete. The goals of the article are ambitious and the article may be valuable for the civil engineers working in the industry. However, from the scientific point of view, the research presented here is not properly designed. Here are some comments regarding this manuscript:
- Table 1. There is no reference mixture used in the research. How the authors are going to establish such an ambitious findings without comparing the results with reference mixture without steel‐slag as the coarse aggregate?
- Are 3 series of concrete enough? In my opinion not – also the range of strengths is small,
- The scale effect was investigated for decades. It seems that the current findings in the article discover the same. However, it is not clear because there is no comparison with other existing models for conventional concrete,
- I suggest to use commonly known water/cement ratio instead of cement/water ratio. Values as 1.76, 2.00, 2.21 seem strange for engineers and researchers,
- it is not clear why the concrete samples were first placed for 24 h in a laboratory with a temperature of 28 ± 5 C and after demolding cured in water with temperature of 25 ± 5 C? This first initial period is very important and keeping the samples in air in such a high temperature will affect the results significantly,
- there are no details about the used materials: steel‐slag, coarse aggregate and fine aggregate. What are they chemical, physical and mechanical properties? Without his knowledge it is not possible to repeat and validate the study in the future.
Finally, in my option the remaining work required to obtain satisfied quality of the manuscript is too large. Thus, I suggest to reject the manuscript at this stage.
Author Response

(The authors gave the same response as above.)

Round 2
Reviewer 1 Report
The paper can be accepted.
Reviewer 3 Report
The manuscript can bez accepted